# Dual Histogram Equalization Algorithm Based on Adaptive Image Correction

Bowen Ye [1,2], Sun Jin [1,2], Bing Li [1,2,*], Shuaiyu Yan [1,2] and Deng Zhang [1,2]

1. School of Mechanical and Automotive Engineering, Guangxi University of Science and Technology, Liuzhou 545006, China; ybw18053456053@163.com (B.Y.); yan7210011999@163.com (S.Y.) 18742062526@163.com (D.Z.)
2. Guangxi Collaborative Innovation Centre for Earthmoving Machinery, Guangxi University of Science and Technology, Liuzhou 545006, China
* Correspondence: gxgxyjxxlb@163.com or libinggxust.edu.cn

**Abstract:** For the visual measurement of moving arm holes in complex working conditions, a histogram equalization algorithm can be used to improve image contrast. To lessen the problems of image brightness shift, image over-enhancement, and gray-level merging that occur with the traditional histogram equalization algorithm, a dual histogram equalization algorithm based on adaptive image correction (AICHE) is proposed. To prevent luminance shifts from occurring during image equalization, the AICHE algorithm protects the average luminance of the input image by improving upon the Otsu algorithm, enabling it to split the histogram. Then, the AICHE algorithm uses the local grayscale correction algorithm to correct the grayscale to prevent the image over-enhancement and gray-level merging problems that arise with the traditional algorithm. It is experimentally verified that the AICHE algorithm can significantly improve the histogram segmentation effect and enhance the contrast and detail information while protecting the average brightness of the input image, and thus the image quality is significantly increased.

**Keywords:** complex working conditions; histogram equalization; Otsu algorithm; machine vision; image enhancement





## 1. Introduction

In the industrial field, machine vision systems applied in practice will inevitably encounter environmental problems (e.g., light, fog, smoke, dust), imaging equipment problems, lighting problems, and other factors that will result in the acquisition of low-quality, low-contrast images, which is not conducive to subsequent image processing, and so image enhancement is necessary. The main methods of image enhancement include histogram equalization, homomorphic filtering [1], Retinex theory-based enhancement algorithm, and deep learning methods. For the image enhancement algorithm of homomorphic filtering, Gong [2] et al. proposed a homomorphic filtering method based on combination and addition in HSV space. However, this work merely improved the underground image data and had certain efficiency flaws. The enhancement method based on Retinex theory has a poor effect on high-brightness images (such as hazy photos), and it produces visible halo phenomena at the intersection of light and dark in the image, which is not conducive to industrial measurement. Deep learning technologies can increase image quality. However, there are still issues with data availability and the generalization of deep learning systems [3].

The histogram equalization method is widely used because it is fast, simple, and effective. Histogram equalization takes the histogram statistics of the pixel values of the input image and then distributes them evenly, which is effective for image enhancement. However, the traditional histogram equalization algorithm [4] can lead to the brightness of the image being offset due to over-stretching, which results in poor enhancement; it can

also lead to a loss of detail information and over-enhancement due to gray-level merging. These image quality problems detract from the success of image processing and hinder the extraction of target information from the image. Therefore, histogram equalization algorithms have been improved through various methods [5–8].

To solve the problem of mean luminance shift, Kim proposed the bi-histogram equalization (BBHE) algorithm [9], which divides the input image histogram into two sub-histograms based on the mean value of the input image histogram, equalizes them separately, and finally merges them. Later on, many other scholars improved such algorithms, and Wang et al. proposed the dualistic sub-image histogram equalization (DSIHE) algorithm [10], which divides the image into two sub-histograms based on the median of the gray level, instead of the mean, and equalizes them separately. The recursive sub-image histogram equalization (RSIHE) algorithm [11] and the recursive mean-separate histogram equalization (RMSHE) algorithm [12] improve upon BBHE and DSIHE, respectively. Chen et al. [13] proposed a bi-histogram equalization algorithm with a "minimum" mean brightness error (i.e., minimum mean brightness error bi-histogram equalization, MMBEBHE), which determines the unique separation point by testing all intensity values and selecting the minimum difference between the average input brightness and the average output brightness. He et al. [14] proposed an infrared image enhancement method combining improved L-C saliency detection and dual-region histogram equalization in order to improve the visual effect of infrared images and highlight the detail information. The foreground and background regions are obtained by adaptive segmentation of the saliency map using the K-means algorithm. Although the K-means algorithm works well when the sample data are dense and the distinction between classes is particularly good, the selection of the K value is difficult to estimate. Blind determination of the K value will lead to inaccurate segmentation results.

The principle of all these methods is to calculate a suitable threshold to split the original histogram and then equalize each histogram separately. These methods can protect the average brightness of the input image, but their limitations are that the segmented sub-histogram is too narrow, leading to poor image enhancement, and the distribution is too wide, so it will contain noise, artifacts, and other defects. To solve the problem of image detail loss, some scholars proposed the local histogram equalization algorithm (AHE) for image contrast enhancement, but the algorithm is complex, has a long running time, and generates a lot of noise and block effects, so it was improved to produce the contrast-limited adaptive histogram equalization (CLAHE) algorithm [15].

In recent years, to mitigate the problem of average brightness change and image detail loss due to gray-level merging in the equalization process, Stark et al. [16] proposed adaptive histogram equalization, the idea of which is to segment the image, perform histogram equalization for each region separately, and finally merge multiple local maps, which can protect certain detail information but also introduce noise. To improve the image over-enhancement problem, Maitra et al. [17] proposed a pre-processing algorithm for pectoral muscle detection and suppression using contrast limited adaptive histogram equalization (ARAN) to enhance the contrast of digital mammograms. Bi-histogram with a plateau limit for digital image enhancement (BHEPL) [18] uses the average of the intensity of each sub-histogram as the platform limit. Aquino-Morínigo et al. [19] proposed a dual Bi-histogram histogram equalization algorithm using two platform limits (BHE2PL). Singh et al. [20] proposed an image enhancement technique using the idea of exposure values, called image enhancement using exposure-based sub-image histogram equalization (ESIHE), which divides the cropped histogram into two parts using a pre-computed exposure threshold. Paul [21] proposed a three-histogram equalization technique for digital image enhancement in the three-platform limit, which uses a separation threshold parameter to initially separate the histogram of the input image into three sub-histograms. Huang [22] proposed an image enhancement strategy—contrast-constrained dynamic quadratic histogram equalization (CLDQHE)—to overcome the drawbacks of over-enhancement and over-smoothing that

exist in traditional histogram equalization methods. Although these algorithms perform well in contrast improvement, they fail to maintain brightness and preserve fine structures.

Hence, this study proposes a dual histogram equalization algorithm based on adaptive image correction (AICHE) for image enhancement in the process of moving arm hole machine vision measurement in complex working conditions. With AICHE, the global histogram is divided into two sub-histograms to solve the problem of mean luminance shift, and then the sub-histograms are corrected in two platform limits to avoid the over-enhancement of the image. Next, to prevent image over-enhancement and gray-level merging problems, grayscale correction is conducted using a local grayscale correction algorithm to perform histogram equalization on the basis of maintaining the average brightness of the input image to improve the image contrast while protecting image detail information.

## 2. Histogram Equalization

The main idea of the histogram equalization algorithm is to extend the probability density function (*PDF*) of the gray levels in the whole image and remap the gray levels of the pixels in the original image. First, the histogram of the original image F is normalized and its cumulative histogram is constructed. The conversion formula is mainly composed of the cumulative distribution function (*CDF*). Then, the cumulative histogram is quantized to the gray level of the output image. The three steps of the algorithm are detailed as follows:

Count the percentage of pixels for each gray value to obtain the *PDF* of the histogram:

$$PDF(i) = \frac{n_i}{n}, i = 0, 1, 2. \cdots k \tag{1}$$

where $i$ is the gray level of the input image, $n$ is the total number of pixels in the input image, and $n_i$ is the total number of pixels in the image with gray level $i$.

Accumulate the *PDF* of each gray level to obtain the *CDF* of the histogram:

$$CDF(i) = \sum_{i=0}^{k} PDF(i) \tag{2}$$

where *CDF* is cumulative distribution function.

Quantize the *CDF* and map it to the output image:

$$F(i) = start + (end - start) \times CDF(i) \tag{3}$$

where *start* and *end* denote the minimum and maximum gray levels of the mapping interval, respectively.

Based on the *CDF*, the traditional histogram equalization algorithm selectively enhances the gray levels that occupy more pixels and extends the distribution range of gray levels. However, it will over-enhance the gray levels with higher frequency, and it will merge the gray levels with fewer pixel points, resulting in the loss of details, which is also the drawback of traditional histogram equalization.

## 3. Proposed AICHE Transformation

In this study, we propose the AICHE algorithm to segment the image into two sub-histograms of target and background by improving upon the Otsu method, and then perform the histogram equalization process separately, which ensures that the average brightness of the original image will not be shifted. Additionally, the algorithm segments the histogram of the image based on the adaptive threshold. This effectively avoids the phenomenon of image over-enhancement and also prevents detail loss to a certain extent. The flowchart of the AICHE algorithm is shown in Figure 1.

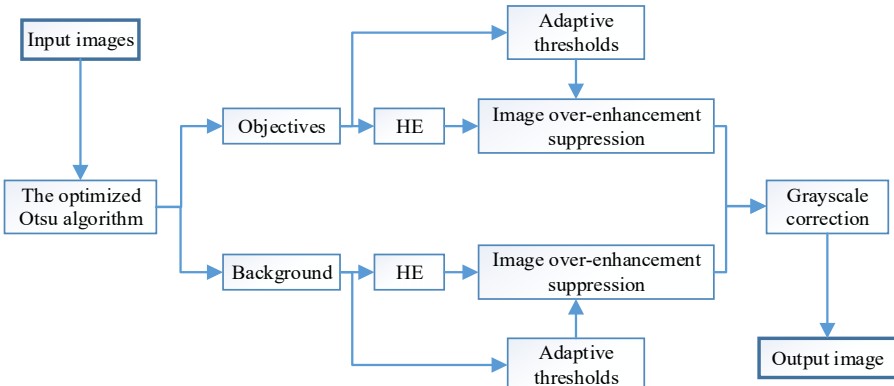

**Figure 1.** Overall flowchart of AICHE algorithm.

### 3.1. Histogram Segmentation

For most images, the distribution of pixel grayscale is not uniform, and the average brightness of the image will be shifted during the equalization process. This can be solved using histogram segmentation. When segmenting the grayscale histogram, if the segmented sub-histogram is too narrow, the equalization effect of the image will be reduced, and if it is too wide, it will lead to excessive enhancement and the loss of details. Therefore, the selection of the threshold value is extremely important, and the improper selection of the threshold value will directly lead to the degradation of the image quality after equalization.

First, suppose a threshold t is the segmentation point, and the image is divided into target region A and background region B according to the gray level, where region A consists of pixels with gray value in the interval $[MIN, t]$, and region B consists of pixels with gray value in the interval $[t + 1, MAX]$. Then, the ratio of class A to class B $q_A(t)$, $q_B(t)$ is

$$q_A(t) = \sum_{i=MIN}^{t} PDF(i), q_B(t) = \sum_{i=T+1}^{MAX} PDF(i) \tag{4}$$

where $MIN$ and $MAX$ denote the initial and termination values of the histogram, respectively, and $P(i)$ denotes the probability that the grayscale value is $i$.

$\mu_A(t)$ and $\mu_B(t)$, can be calculated as follows:

$$\mu_A(t) = \sum_{i=MIN}^{t} \frac{iPDF(i)}{q_A(t)}, \mu_B(t) = \sum_{i=T+1}^{MAX} \frac{iPDF(i)}{q_B(t)} \tag{5}$$

where $\mu_A(t)$ and $\mu_B(t)$ denote the probabilities of class A and class B, respectively.

Then, the $\mu$ can be calculated as follows:

$$\mu = q_A(t)\mu_A(t) + q_B(t)\mu_B(t) \tag{6}$$

where $\mu$ is the average grayscale of the input image.

The inter-class variance $\sigma_T^2$ is defined as

$$\sigma_T^2 = q_A(t)[\mu_A(t) - \mu]^2 + q_B(t)[\mu_B(t) - \mu]^2 \tag{7}$$

The traditional Otsu algorithm is simple, convenient, and not affected by the brightness of the image. It sets the threshold at which the variance between the target and background grayscale reaches its maximum value as the optimal segmentation threshold: $K_{Otsu} = \arg_t \min \sigma_T^2$. The smaller the distance between each pixel in two regions and the class center, the better the pixel cohesion in each region. The traditional Otsu algorithm

is less effective in segmentation because it does not consider pixel spatial correlation. To measure the goodness of pixel cohesion, $d^2(t)$ is assumed and calculated as follows:

$$d^2(t) = (\mu_A(t) - \mu_B(t))^2 \tag{8}$$

where $d^2(t)$ is a distance metric. $\sigma_A^2$ and $\sigma_B^2$ are calculated as follows:

$$\sigma_A^2(t) = \frac{1}{q_A(t)} \sum_{i=MIN}^{t} (i - \mu_A(t))^2 P(i) \tag{9}$$

$$\sigma_B^2(t) = \frac{1}{q_B(t)} \sum_{i=t+1}^{MAX} (i - \mu_B(t))^2 P(i) \tag{10}$$

where $\sigma_A^2$ and $\sigma_B^2$ denote the mean variance values of the target and background regions, respectively.

Obviously, the smaller the average variances $\sigma_A^2$ and $\sigma_B^2$, the better the segmentation effect; on this basis, a new threshold-finding formula $G(t)$ is obtained.

$$G(t) = \frac{q_A(t)q_B(t)d^2(t)}{\sigma_A^2(t) + \sigma_B^2(t)} = \frac{q_A(t)q_B(t)(\mu_A(t) - \mu_B(t))^2}{\sigma_A^2(t) + \sigma_B^2(t)} \tag{11}$$

The corresponding $t$ when $G(t)$ takes the maximum value is the optimal threshold. Therefore, $K_{out}$ is obtained as follows:

$$K_{out} = \arg_t \max G(t) \tag{12}$$

where $K_{out}$ is the optimal threshold value.

According to threshold $K_{out}$, the histogram is divided into two sub-histograms, where the first part is defined as $i \in [0 : K_{out}]$ and the second part is defined as $i \in (K_{out} + 1 : L)$.

### 3.2. Adaptive Local Grayscale Correction

The input image is divided into two sub-histograms according to the algorithm above, and histogram equalization is performed on each of the two sub-histograms to improve the image brightness offset. However, a new histogram assignment algorithm is used in this study to solve the image over-enhancement and gray level merging problems. The algorithm is mainly divided into two parts: image over-enhancement suppression and local gray level correction.

### 3.2.1. Image Over-Enhancement Suppression

In the equalization process, the gray levels with higher frequencies appear to be over-enhanced, whereas the gray levels with lower frequencies are merged, leading to a loss of image details. Therefore, the AICHE algorithm suppresses the over-enhanced gray levels by setting a threshold T for each of the two sub-histograms. The procedure is as follows.

1. First, let the input image be $F$, and obtain the sets $F_A$ and $F_B$ of non-zero cells in the two sub-histograms.

$$\begin{cases} F_A = \{F(i)|F(i) \neq 0\}, i \in [MIN, K_{out}] \\ F_B = \{F(i)|F(i) \neq 0\}, i \in [K_{out} + 1, MAX] \end{cases} \tag{13}$$

where $i$ is the gray level of the image, and $F_A$ and $F_B$ denote non-zero cells in the two sub-histograms, respectively.

2.  The one-dimensional median filtering of $F_A$ and $F_B$ is performed, and the segmentation thresholds $T_A$ and $T_B$ of the two sub-histograms are calculated as follows.

$$\begin{cases} T_A = T_{MA} \times \frac{K_{out} - MIN}{MAX - MIN} \\ T_B = T_{MB} \times \frac{MAX - K_{out}}{MAX - MIN} \end{cases} \tag{14}$$

where $T_{MA}$ and $T_{MB}$ denote the peaks of the two sub-histograms, respectively.

3.  The image $P_S$ is obtained by independently equalizing the two sub-histograms according to Equations (1)–(3), and the equalization equation is as follows.

$$P_S(i) = \begin{cases} MIN + \frac{(K_{out} - MIN) \times \sum\limits_{i=MIN+1}^{K_{out}} PDF(i)}{NA}, i \leq K_{out} \\ (K_{out} + 1) + \frac{[MAX - (K_{out}+1)] \times \sum\limits_{i=K_{out}}^{MAX} PDF(i)}{NB}, i > K_{out} \end{cases}$$

$$= \begin{cases} MIN + \frac{(K_{out} - MIN) \times \sum\limits_{i=MIN+1}^{K_{out}} \frac{n_i}{n}}{NA}, i \leq K_{out} \\ (K_{out} + 1) + \frac{(MAX - K_{out} - 1) \times \sum\limits_{i=MIN+1}^{K_{out}} \frac{n_i}{n}}{NB}, i > K_{out} \end{cases} \tag{15}$$

where $n_i$ is the total number of pixels in the image at gray level $i$, $P_S(i)$ is the histogram after equalization at gray level $i$, and $NA$ and $NB$ are the total numbers of gray levels in region A and region B, respectively.

4.  After cropping the balanced histogram according to Equation (16), the image $P_T$ is obtained.

$$P_T(i) = \begin{cases} T_A, i \leq K_{out} \cap P_S(i) \geq T_A \\ T_B, i > K_{out} \cap P_S(i) \geq T_B \\ P_S(i) \end{cases} \tag{16}$$

where $P_T(i)$ indicates the cropped histogram with gray level $i$.

### 3.2.2. Local Gray Level Correction

To solve the problem that the gray levels will be merged after equalization, the AICHE algorithm corrects the image after equalization. First, the gradient value is obtained by convolving the input image and the equalized image with the Sobel operator to find the location where the gradient value is obviously reduced. Second, the gray value is modified with reference to the original image to enhance the local gradient value to protect the image detail information. The specific process is as follows.

1.  The gradient matrices $D_{in}$ and $D_{HE}$ of the input image and the equalized image are obtained by convolving the images $F$ and $P_T$ with Sobel operators in four directions. The gradient matrix convolution is calculated as follows:

$$D_{in} = \sqrt{(D_{0°} * F)^2 + (D_{180°} * F)^2 + (D_{45°} * F)^2 + (D_{135°} * F)^2}$$
$$D_{HE} = \sqrt{(D_{0°} * P_T)^2 + (D_{180°} * P_T)^2 + (D_{45°} * P_T)^2 + (D_{135°} * P_T)^2} \tag{17}$$

where $D_{0°}, D_{45°}, D_{135°}, D_{180°}$ denote the convolution factors in the four directions of $0°, 45°, 135°,$ and $180°$, respectively. The four convolution factors are

$$D_{0°} = \begin{bmatrix} -1 & 0 & 1 \\ -2 & 0 & 2 \\ -1 & 0 & 1 \end{bmatrix}; D_{180°} = \begin{bmatrix} -1 & -2 & -1 \\ 0 & 0 & 0 \\ 1 & 2 & 1 \end{bmatrix};$$

$$D_{45°} = \begin{bmatrix} 2 & 1 & 0 \\ 1 & 0 & -1 \\ 0 & -1 & -2 \end{bmatrix}; D_{135°} = \begin{bmatrix} 0 & -1 & -2 \\ 1 & 0 & -1 \\ 2 & 1 & 0 \end{bmatrix}; \tag{18}$$

2. Local grayscale correction of the image $P_T$ is conducted according to Equation (19) to enhance the local information of the image.

$$x_{out}(i,j) = \begin{cases} x_{main}^{HE}(i,j) + (x_{in}(i,j) - x_{main}(i,j)), D_{HE}(i,j) < D_{in}(i,j) \\ x_{HE}(i,j), D_{HE}(i.j) \geq D_{in}(i,j) \end{cases} \tag{19}$$

where $x_{out}(i,j)$ is the grayscale value of the center pixel of the output image, $x_{in}(i,j)$ and $x_{HE}(i,j)$ denote the center pixels of image F and image $P_T$, respectively, and $x_{main}(i,j)$ and $x_{main}^{HE}(i,j)$ are the grayscale averages of each pixel in a $5 \times 5$ window centered at $(i,j)$ in the input image and the equalized image, respectively.

3. The final image is the output.

Figure 2 shows the effect of image processing and its grayscale histogram during the process of the HE algorithm and AICHE algorithm. It can be seen that although the HE algorithm can improve the image contrast, the image is overexposed due to image stretching. And, after histogram segmentation, the average brightness of the image is protected, but at this point there is still the problem of gray level merging and the loss of image details. After the adaptive local gray level correction of the image, the average brightness of the input image is protected while the contrast and detail information are enhanced, and the image quality is significantly improved.

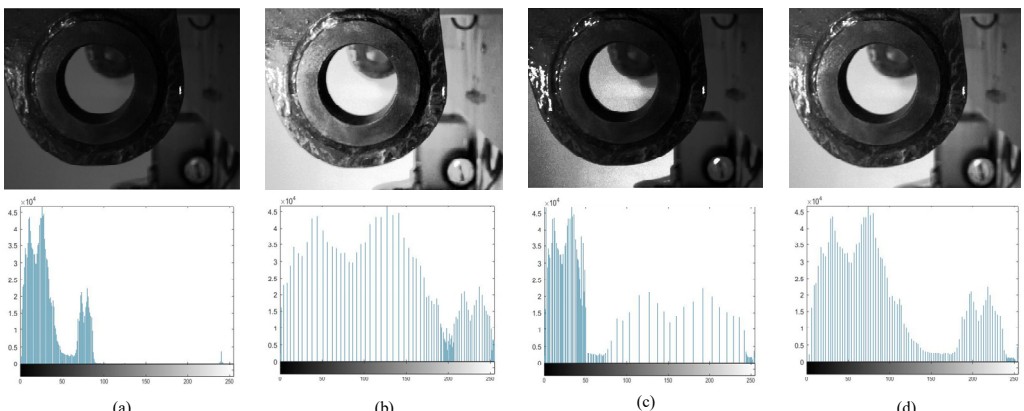

| (a) | (b) | (c) | (d) |

**Figure 2.** Image enhancement effect of AICHE algorithm: (**a**) original image; (**b**) effect of HE algorithm; (**c**) effect of histogram segmentation; and (**d**) effect of adaptive local grayscale correction.

## 4. Analysis of Algorithm Results

### 4.1. Improved Image Segmentation Effect of Otsu Algorithm

Figure 3 shows the comparison between the improved Otsu algorithm and other image segmentation algorithms in three scenarios. Unlike the traditional Otsu and K-means algorithms, the improved Otsu algorithm can segment the image reasonably well to obtain a more complete moving arm profile. The improved Otsu algorithm segmentation can show more details of the image and optimize the segmentation effect.

### 4.2. AICHE Algorithm Effect

To demonstrate the effectiveness of the AICHE algorithm, Figures 4–13 simulate the environment of insufficient light, fog, and smoke, and compare the image enhancement effects of seven histogram equalization algorithms with those of the AICHE algorithm. These include the classical algorithms HE, BBHE, and CLAHE and several more advanced algorithms, BHEPL, RSIHE, ESIHE, and MEBEBHE.

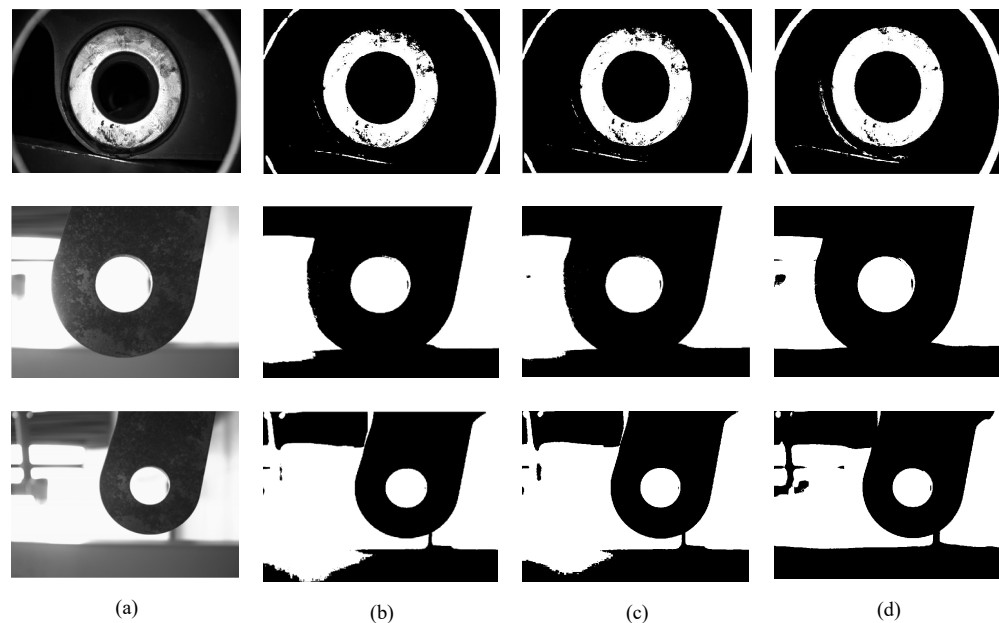

(a)           (b)           (c)           (d)

**Figure 3.** Comparison of segmentation effect between improved Otsu algorithm and other image segmentation algorithms: (**a**) original image; (**b**) segmentation effect of traditional Otsu algorithm method; (**c**) segmentation effect of the K-means algorithm method; and (**d**) segmentation effect of improved Otsu algorithm.

### 4.3. Objective Evaluation Indicators

Four objective evaluation indicators are selected in this study, which are detailed below.

#### 4.3.1. Structure Similarity Index Measure

Structure similarity index measure (SSIM) is a metric used to compare the similarity of two images. The SSIM value is mainly based on three characteristics: structure, luminance, and contrast. Luminance is measured by the average gray value; contrast is measured by the gray standard deviation; and structure is measured by the correlation coefficient. The calculation method is as follows:

$$\mu = \frac{1}{N}\sum_{i=1}^{N} x_i \tag{20}$$

$$\sigma = \left(\frac{1}{N-1}\sum_{i=1}^{N}(x_i - \mu)^2\right)^{1/2} \tag{21}$$

where $\mu$ is the average gray value, $\sigma$ is the gray standard deviation, and $C$ is the correlation coefficient.

*SSIM* is consistent with human visual characteristics in evaluating image quality. Its value falls in the range of [0, 1], where a higher value indicates a stronger similarity between the two images, reflecting higher image quality. Its calculation formula is as follows:

$$SSIM(X,Y) = \frac{(2\mu_x\mu_y + C_1)(2\sigma_{xy} + C_2)}{(\mu_x^2 + \mu_y^2 + C_1)(\sigma_x^2 + \sigma_y^2 + C_2)} \tag{22}$$

where $X$ and $Y$ denote the input image and output image, respectively; $\sigma_x$ and $\sigma_y$ are the standard deviations of image $X$ and $Y$, respectively; $\mu_x$ and $\mu_y$ are the grayscale averages of image $X$ and image $Y$, respectively; $\sigma_{xy}$ is the covariance of the two images; and $C_1$ and $C_2$ are the correlation coefficients.

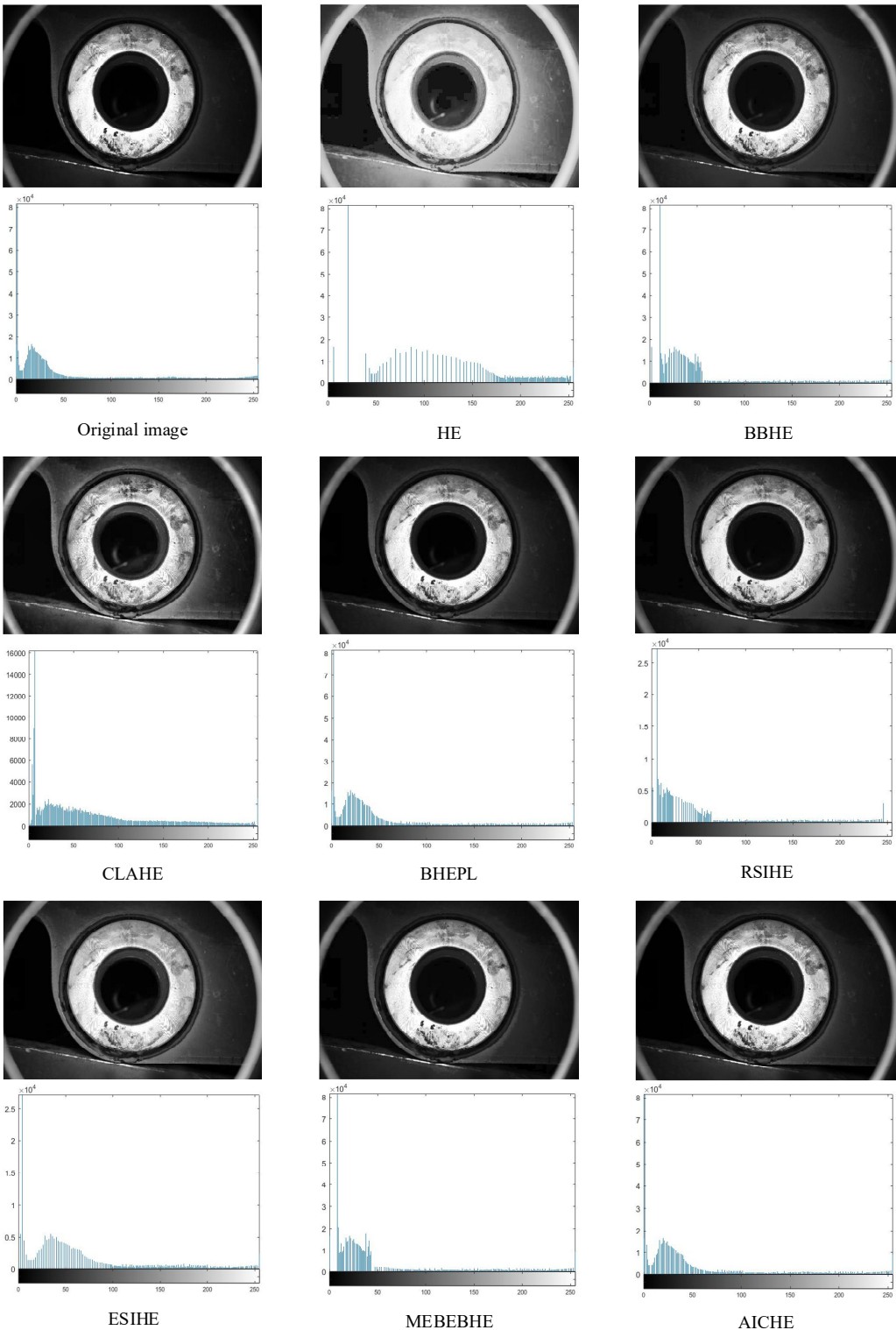

**Figure 4.** The simulation results (**above**) of the 'scene 1' image are presented along with its corresponding histogram (**below**).

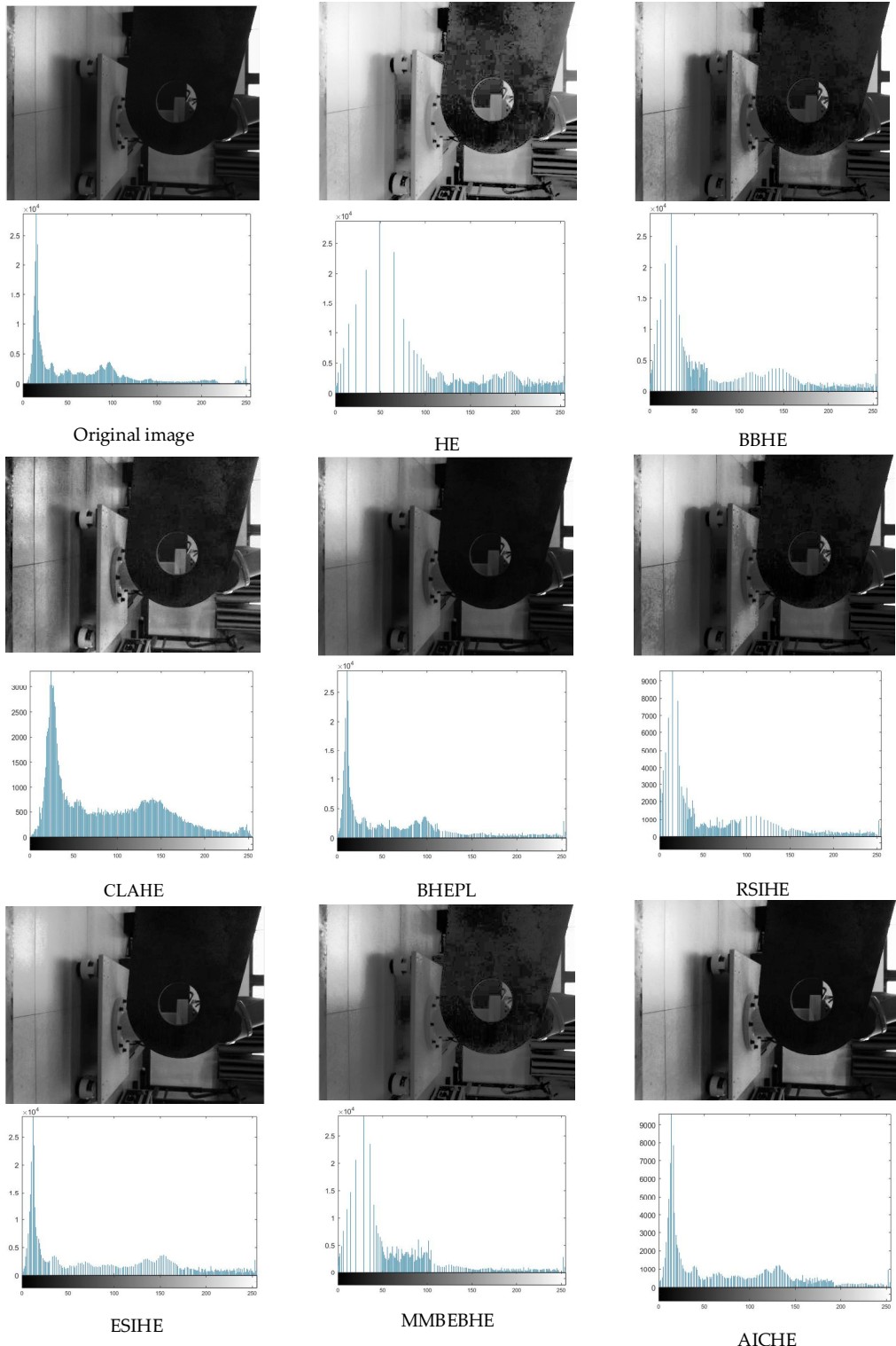

**Figure 5.** The simulation results (**above**) of the 'scene 2' image are presented along with its corresponding histogram (**below**).

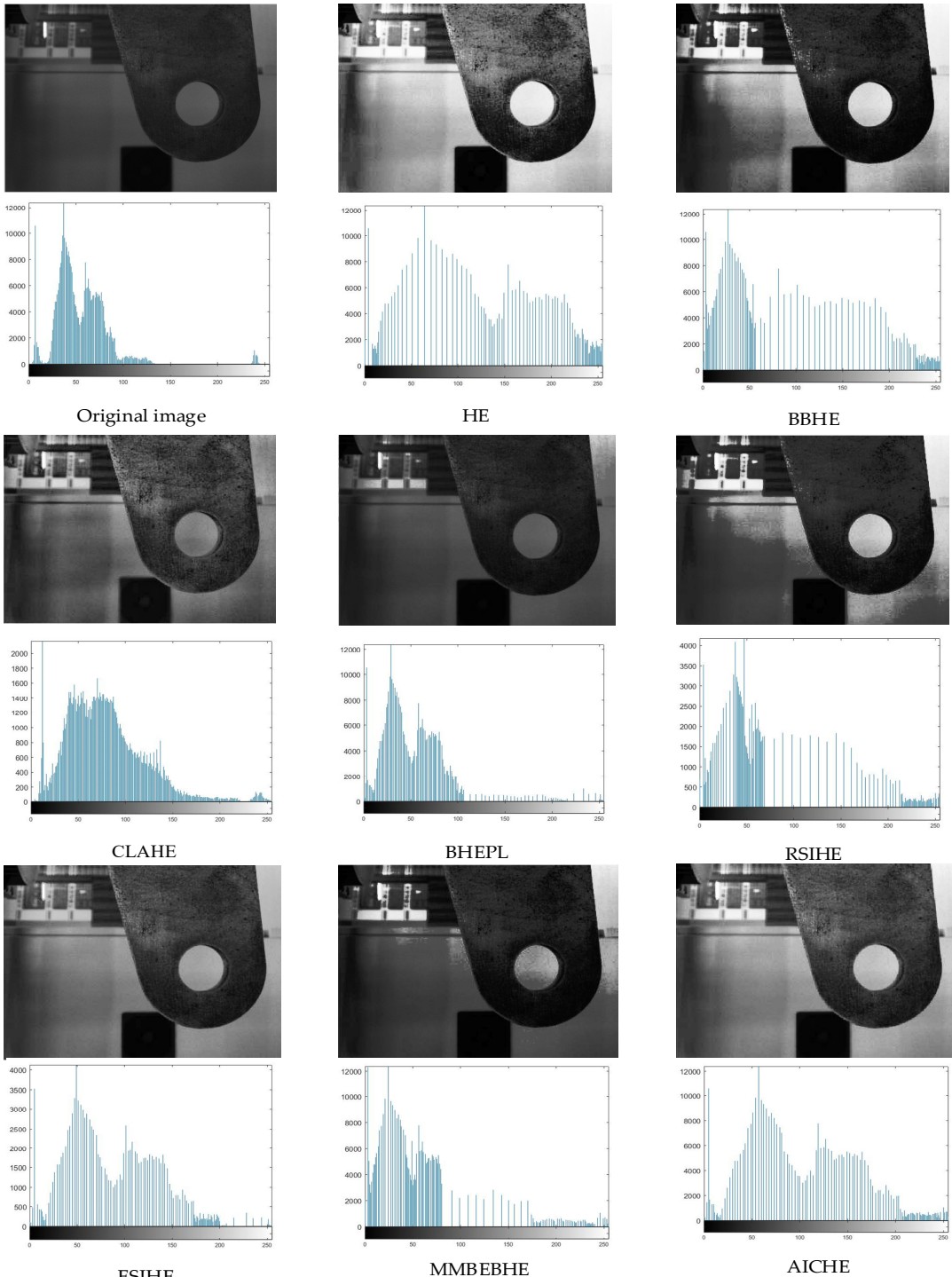

**Figure 6.** The simulation results (**above**) of the 'scene 3' image are presented along with its corresponding histogram (**below**).

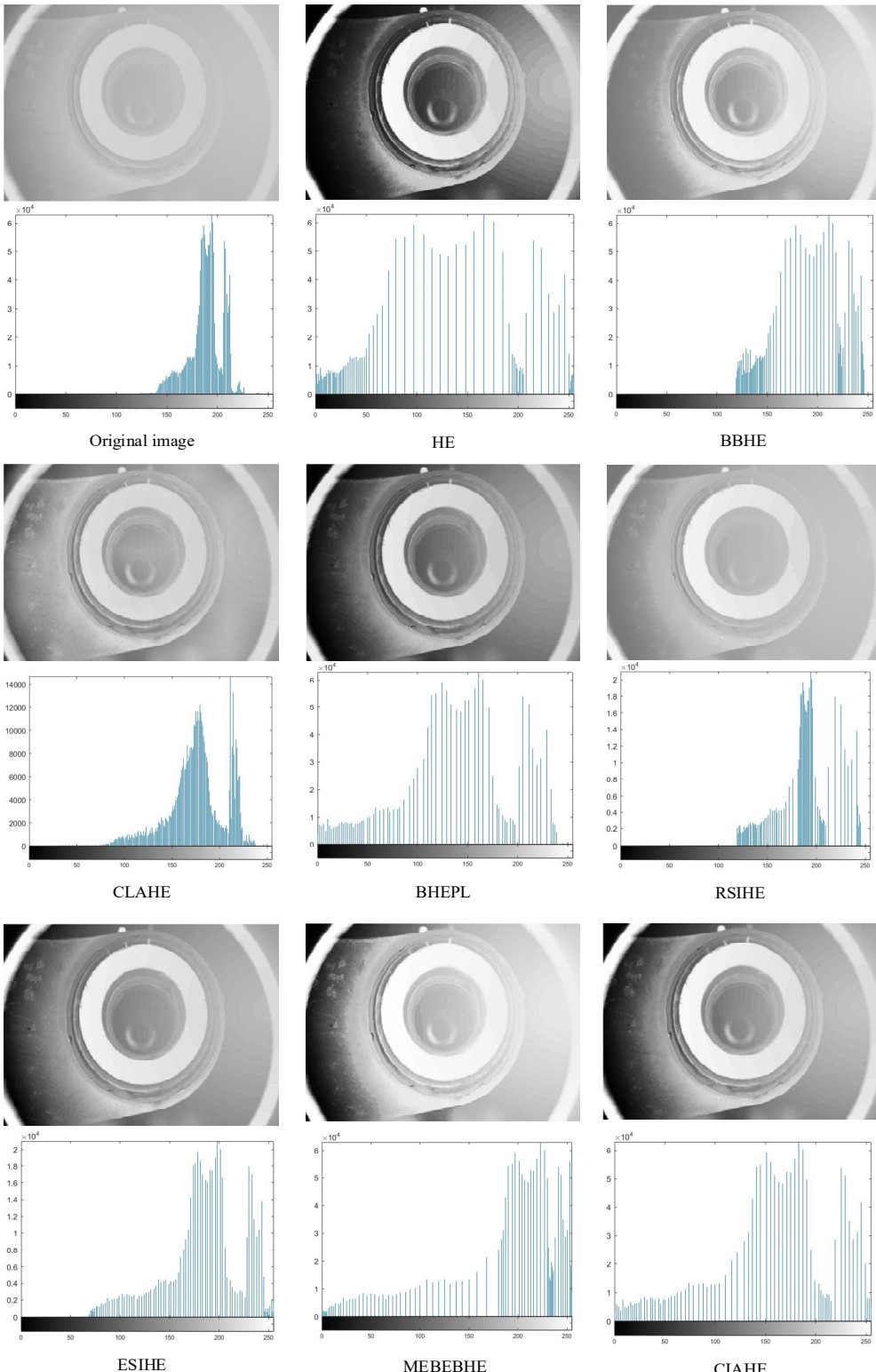

**Figure 7.** The simulation results (**above**) of the 'scene 4' image are presented along with its corresponding histogram (**below**).

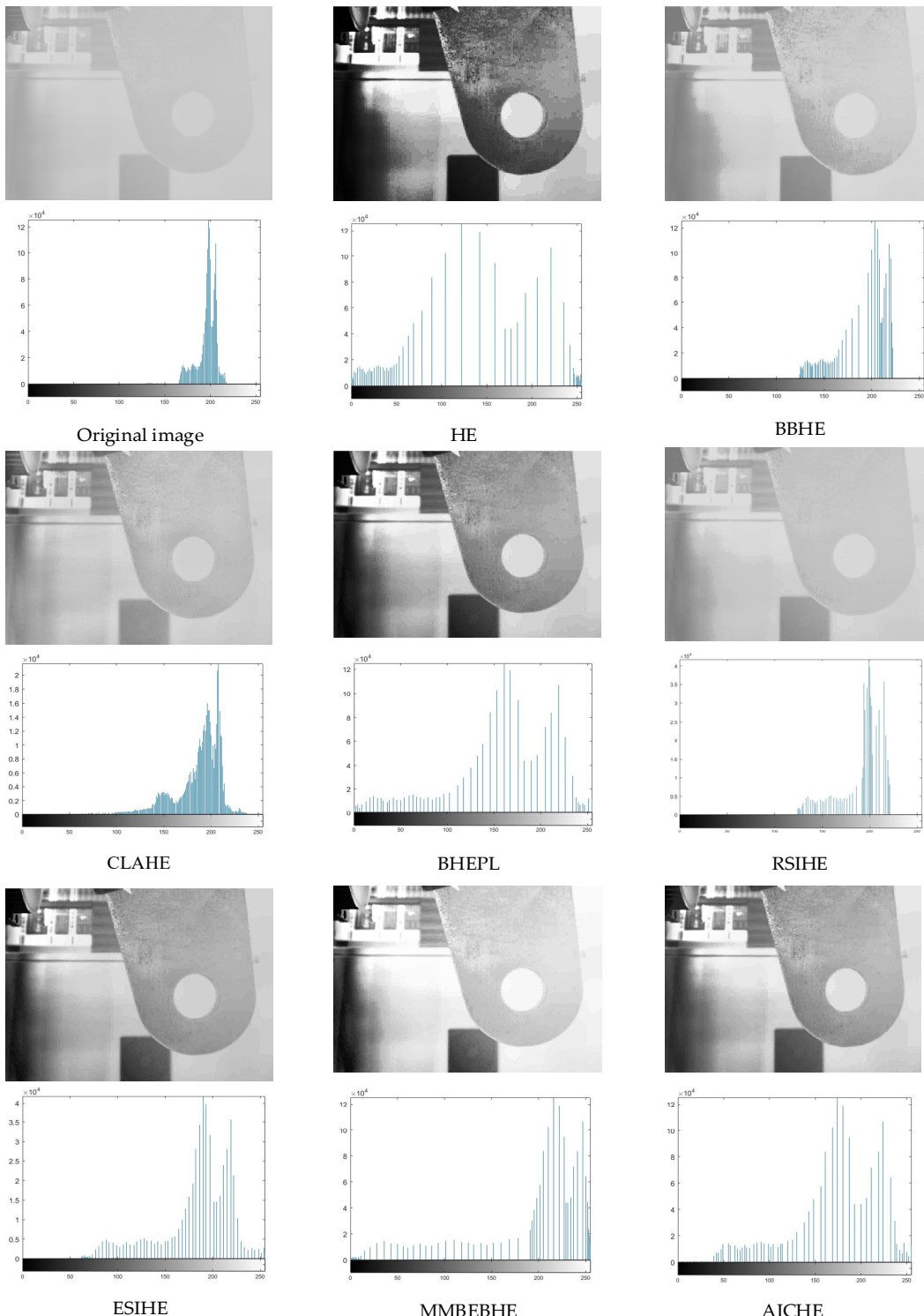

**Figure 8.** The simulation results (**above**) of the 'scene 5' image are presented along with its corresponding histogram (**below**).

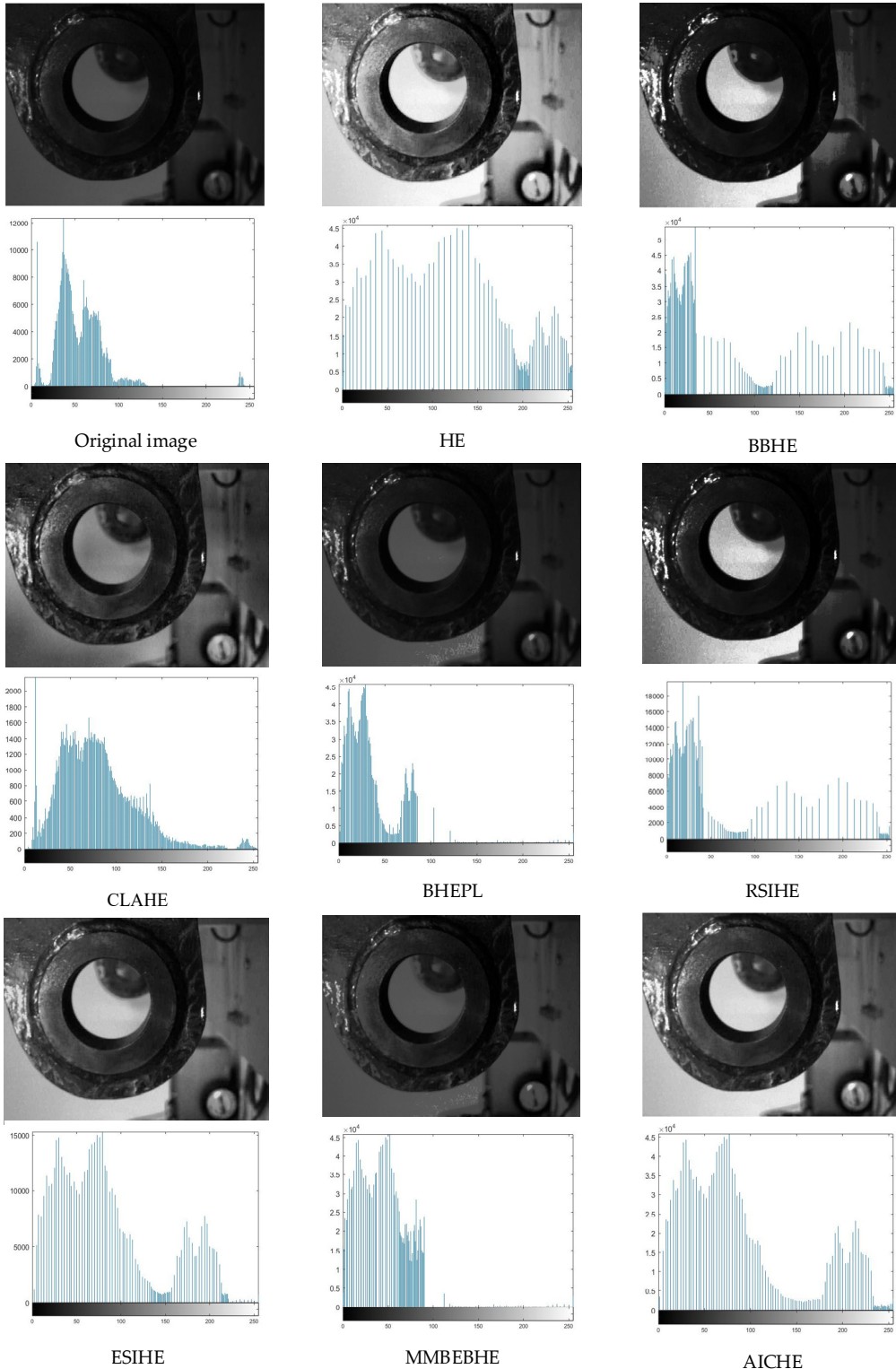

**Figure 9.** The simulation results (**above**) of the 'scene 6' image are presented along with its corresponding histogram (**below**).

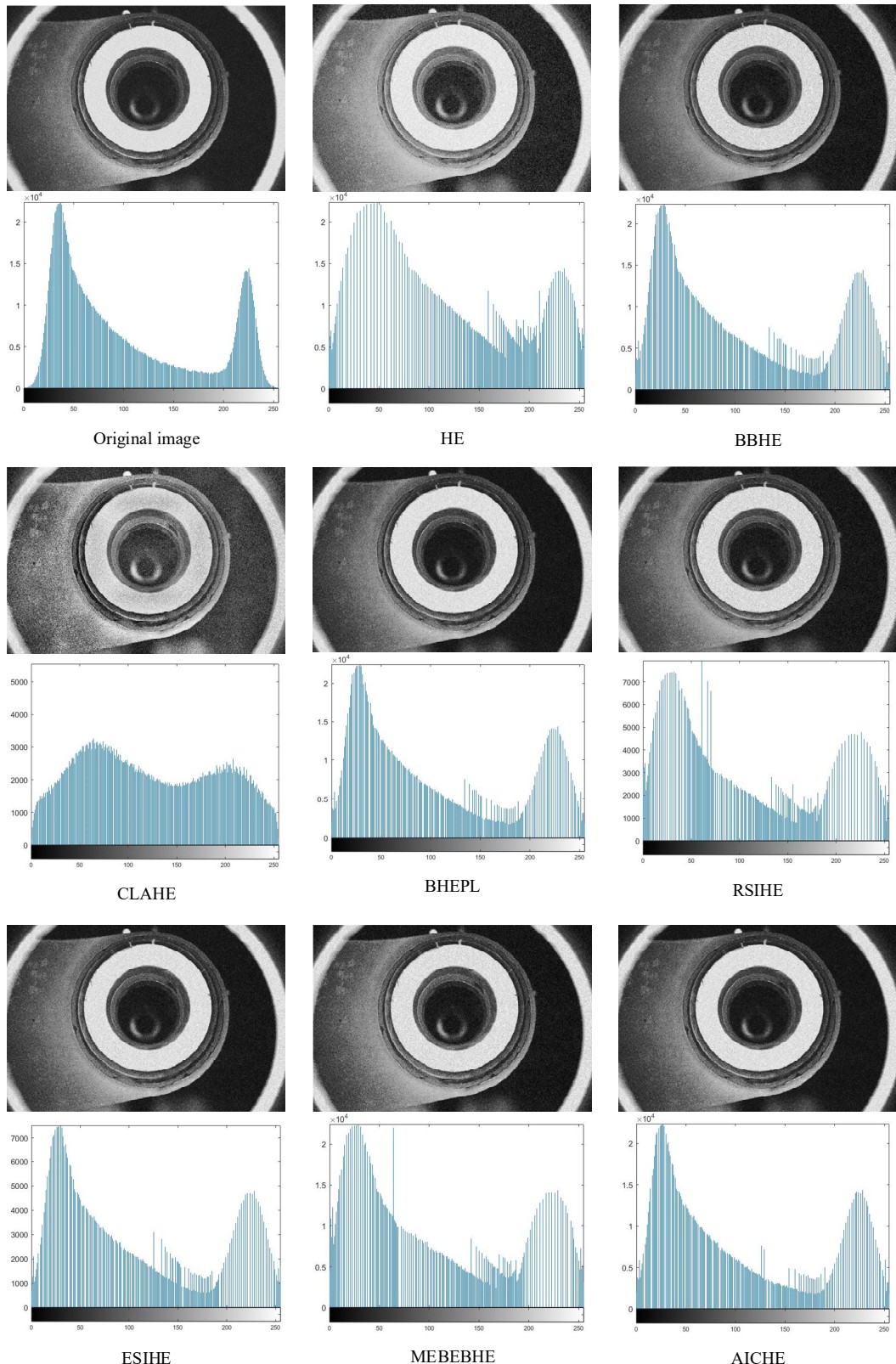

**Figure 10.** The simulation results (**above**) of the 'scene 7' image are presented along with its corresponding histogram (**below**).

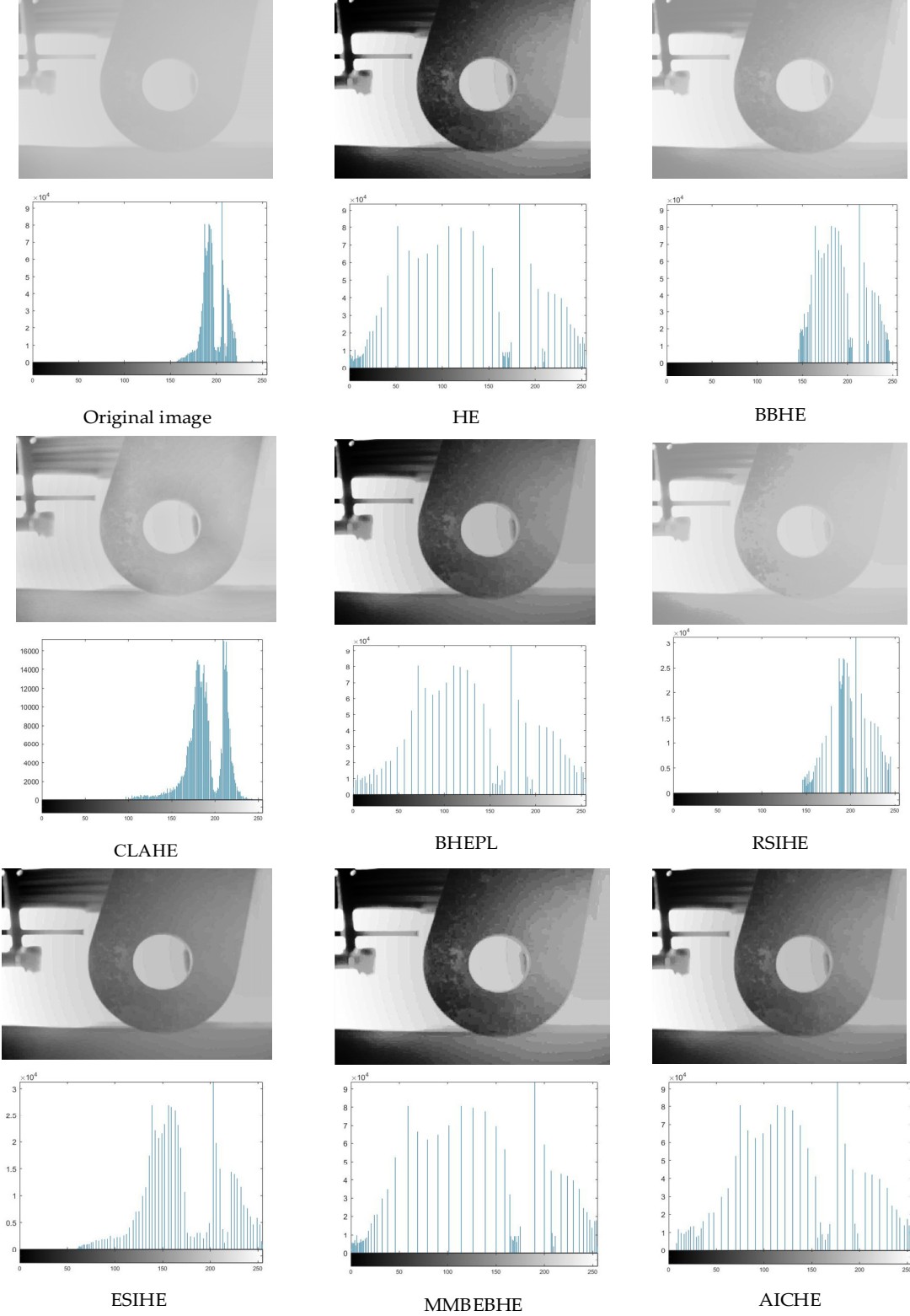

**Figure 11.** The simulation results (**above**) of the 'scene 8' image are presented along with its corresponding histogram (**below**).

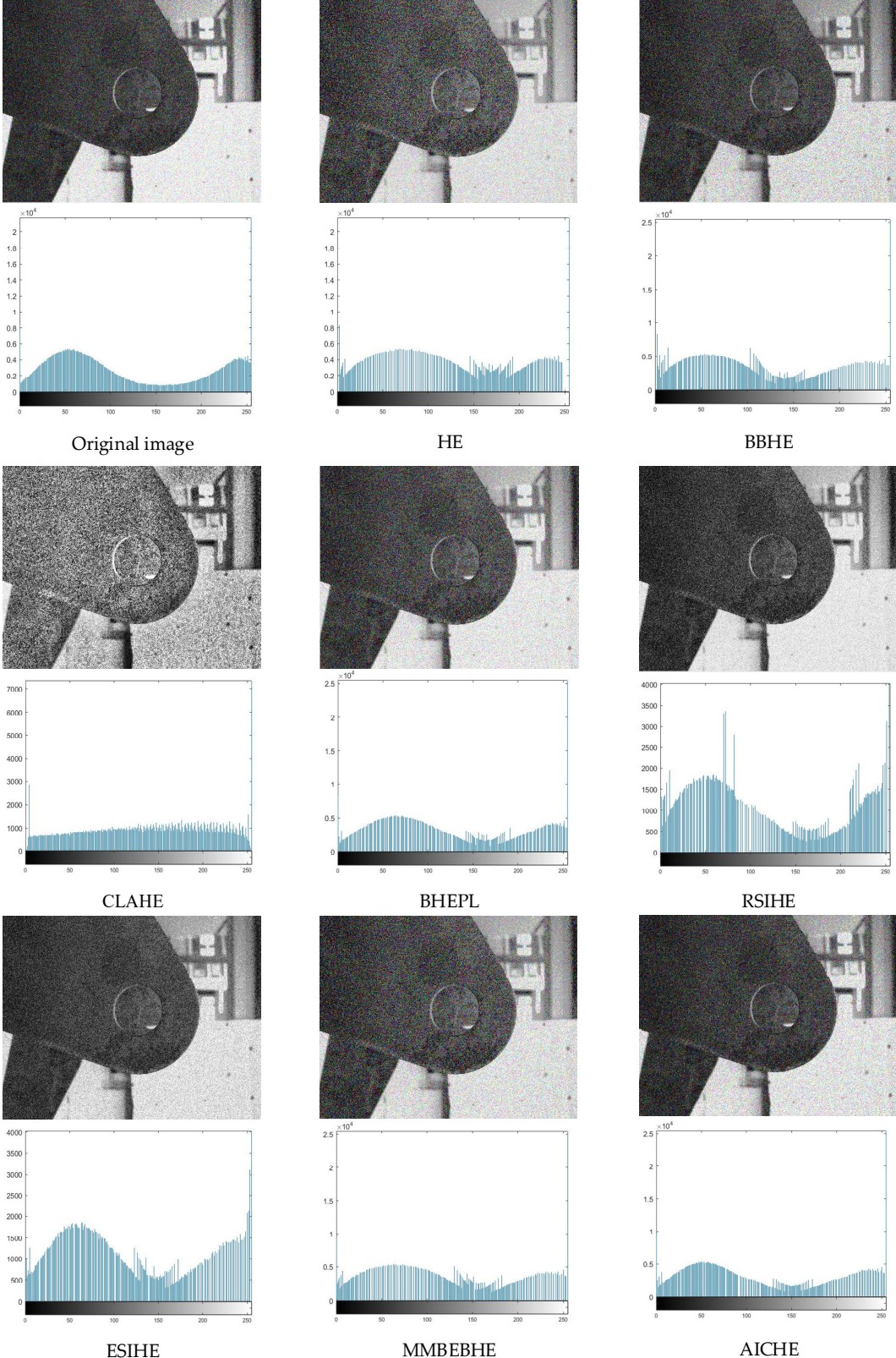

**Figure 12.** The simulation results (**above**) of the 'scene 9' image are presented along with its corresponding histogram (**below**).

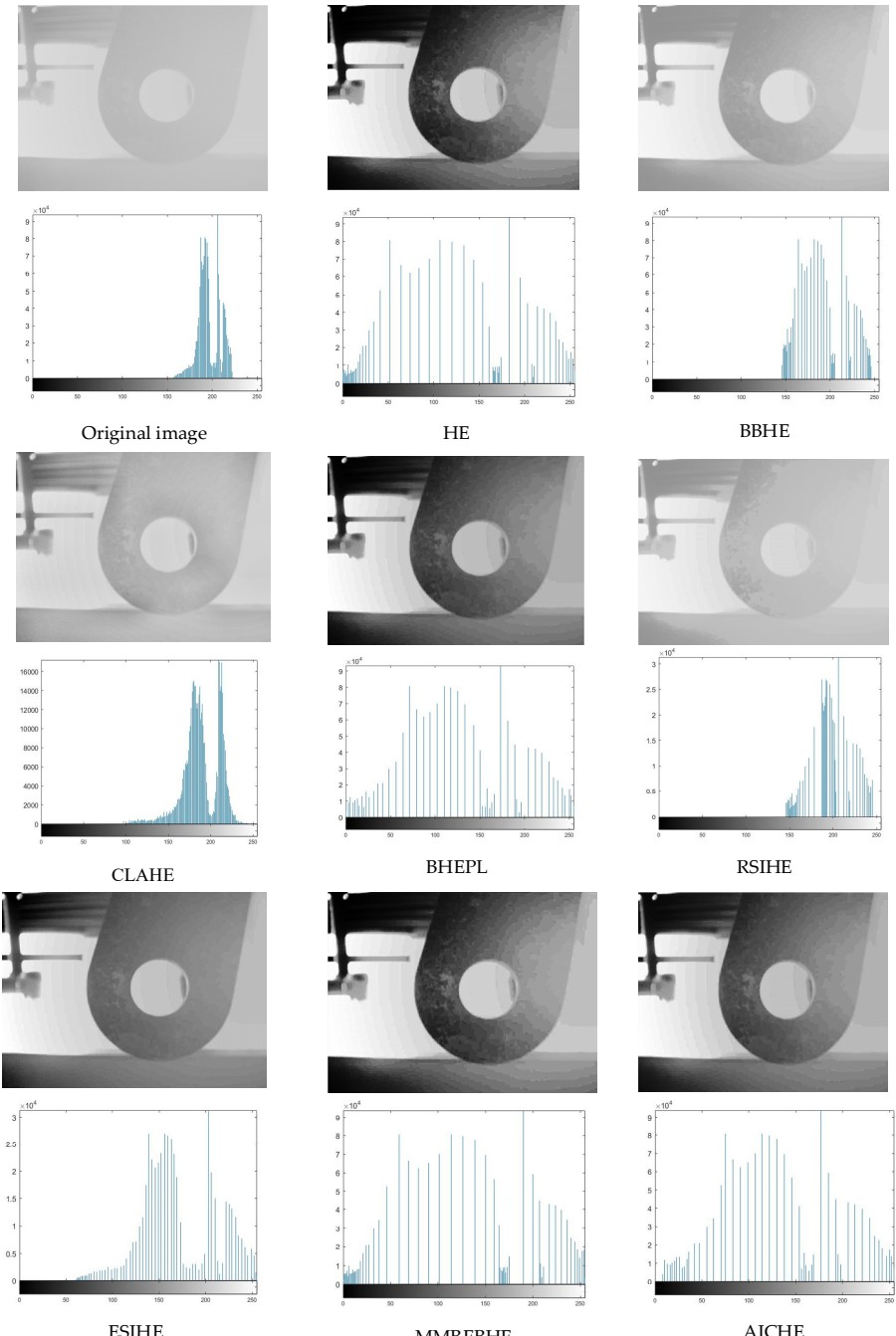

**Figure 13.** The simulation results (**above**) of the 'scene 10' image are presented along with its corresponding histogram (**below**).

4.3.2. Peak Signal-to-Noise Ratio

Peak signal-to-noise ratio (*PSNR*) can be used to compare the contrast enhancement effect of images. PSNR is a measure of image quality based on the definition of mean square error (*MSE*), which expresses the average of the differences between two images at each pixel point and is calculated as follows:

$$MSE = \frac{\sum\limits_{i=1}^{N}\sum\limits_{j=1}^{N}|X(i,j) - Y(i,j)|^2}{N} \tag{23}$$

*PSNR* is improved on the basis of *MSE* and its value is greater than zero. The larger the value, the less distortion in the output image, the higher the contrast, and the more obvious the enhancement effect; it is calculated as follows:

$$PSNR = 10 \log_{10}[\frac{(2^N - 1)^2}{MSE}] \qquad (24)$$

where $N$ is the total number of pixels; $X(i, j)$ and $Y(i, j)$ denote the input image and output image, respectively; and *MSE* is the mean square error.

### 4.3.3. Absolute Mean Brightness Error

The absolute average brightness error (*AMBE*) indicates the absolute difference between the average brightness of the input image and the resulting image, and it is used to measure the performance in maintaining the original brightness. ASME is a value greater than zero. The smaller the *AMBE* value, the better the light preservation effect. It is calculated as follows:

$$AMBE = |Q(X) - Q(Y)| \qquad (25)$$

where $X$ and $Y$ denote the input image and the resultant image, respectively, and $Q(X)$ and $Q(Y)$ are the average brightness values of the input image and the resultant image, respectively.

### 4.3.4. Information Entropy

Information entropy (*E*) is used to measure the information richness of an image, which is greater than zero. The larger the value of *E*, the more information and details are present in the image. However, a large value of *E* also indicates significant noise in the image. It is calculated as follows:

$$E = -\sum_{i=0}^{255} p_i \log p_i \qquad (26)$$

where $p_i$ denotes the proportion of pixels with gray value $i$ in the image.

### *4.4. Evaluation Results*

In this study, the performance of each algorithm is measured using image evaluation metrics such as SSIM, E, PSNR, AMBE, and time.

As can be seen in Figures 4–13, the HE algorithm shows an obvious phenomenon of image brightness change and detail loss. Under low light conditions, the BBHE, RSIHE, and MMBEBHE algorithms can protect the average brightness but will lead to uneven histogram balance due to unreasonable histogram segmentation, which is less effective for detail processing and will cause local area distortion. CLAHE can protect the image details but will introduce a large amount of noise, especially in Figures 10 and 13 where the image visual effect is significantly reduced. The AICHE algorithm improves the image contrast under the condition of protecting the average brightness of the image; it does not introduce excessive noise, and, to a certain extent, it retains the original shape of the original histogram.

From Tables 1 and 2, the AICHE algorithm has the highest PSNR value and the SSIM value is closest to 1.The image information entropy in Table 3 demonstrates the richness of details in the image. The information entropy of the image processed by CLAHE obviously exceeds that of the original image, which indicates that noise is introduced in the image and produces a block effect. Except for the CLAHE algorithm, the AICHE algorithm has the highest information entropy value, indicating that the gray level merging phenomenon is effectively avoided in the equalization process, which can protect the image information. ESIHE cannot maintain the average image brightness well, which is also clearly reflected by the AMBE values in Table 4.

**Table 1.** SSIM of different algorithms.

| Image | HE | BBHE | CLAHE | BPLHE | RSIHE | ESIHE | MMBEBHE | AICHE |
|---|---|---|---|---|---|---|---|---|
| Scene 1 | 0.35355 | 0.81761 | 0.65532 | 0.96649 | 0.89335 | 0.76816 | 0.87611 | 0.97822 |
| Scene 2 | 0.56202 | 0.82984 | 0.66354 | 0.86973 | 0.89335 | 0.90575 | 0.83181 | 0.91081 |
| Scene 3 | 0.40267 | 0.83725 | 0.47892 | 0.88366 | 0.87225 | 0.70318 | 0.87699 | 0.89607 |
| Scene 4 | 0.65454 | 0.94258 | 0.65375 | 0.77875 | 0.95223 | 0.87196 | 0.86033 | 0.95223 |
| Scene 5 | 0.58603 | 0.84615 | 0.63251 | 0.75727 | 0.88304 | 0.90293 | 0.84839 | 0.92898 |
| Scene 6 | 0.57518 | 0.72178 | 0.63395 | 0.85655 | 0.81686 | 0.83812 | 0.85081 | 0.89026 |
| Scene 7 | 0.79064 | 0.89123 | 0.63185 | 0.95572 | 0.89463 | 0.9434 | 0.89365 | 0.95629 |
| Scene 8 | 0.70979 | 0.88585 | 0.76452 | 0.77475 | 0.81018 | 0.89923 | 0.71782 | 0.91879 |
| Scene 9 | 0.90019 | 0.94821 | 0.91324 | 0.96226 | 0.97794 | 0.96686 | 0.92514 | 0.97873 |
| Scene 10 | 0.68766 | 0.77379 | 0.69663 | 0.83925 | 0.78948 | 0.83556 | 0.70165 | 0.85255 |
| Average value | 0.62222 | 0.84942 | 0.67242 | 0.86444 | 0.87833 | 0.863515 | 0.83827 | **0.926293** |
| Standard deviation | 0.16538 | 0.07077 | 0.11052 | 0.07907 | 0.06007 | 0.08025 | 0.07265 | 0.04078 |

Note: The top averages produced by the compared algorithms are marked in bold font.

**Table 2.** PSNR of different algorithms.

| Image | HE | BBHE | CLAHE | BPLHE | RSIHE | ESIHE | MMBEBHE | AICHE |
|---|---|---|---|---|---|---|---|---|
| Scene 1 | 9.7051 | 28.2200 | 13.4224 | 32.8191 | 27.2202 | 20.3642 | 30.8145 | 33.1691 |
| Scene 2 | 11.0532 | 18.8172 | 12.1940 | 17.7522 | 18.7340 | 22.4839 | 23.4246 | 31.9793 |
| Scene 3 | 7.6270 | 13.9420 | 11.5169 | 22.6969 | 10.3618 | 12.1808 | 23.5157 | 27.3033 |
| Scene 4 | 9.6131 | 13.2657 | 17.2237 | 11.5263 | 21.4071 | 20.5795 | 14.7804 | 20.5795 |
| Scene 5 | 8.7144 | 14.1542 | 17.9418 | 11.8057 | 21.8398 | 18.4647 | 14.3072 | 23.1845 |
| Scene 6 | 9.2463 | 12.6208 | 10.5965 | 11.0304 | 10.7381 | 12.3334 | 9.8361 | 12.7580 |
| Scene 7 | 17.4424 | 27.7650 | 11.7832 | 30.9384 | 22.9990 | 30.1157 | 26.4948 | 31.3333 |
| Scene 8 | 9.0751 | 25.2620 | 19.9886 | 10.4082 | 25.1913 | 16.6951 | 9.2755 | 21.3871 |
| Scene 9 | 21.2349 | 24.7633 | 6.8659 | 26.1318 | 21.5225 | 27.5689 | 23.6395 | 29.8354 |
| Scene 10 | 8.813 | 23.4973 | 20.3784 | 9.4208 | 22.6432 | 16.3077 | 9.1325 | 25.8520 |
| Average value | 11.2524 | 20.2307 | 14.1911 | 18.4529 | 20.2657 | 19.7093 | 18.5221 | **25.7381** |
| Standard deviation | 4.43957 | 6.34223 | 4.46301 | 9.02647 | 5.60078 | 5.89009 | 7.96312 | 6.37072 |

Note: The top averages produced by the compared algorithms are marked in bold font.

**Table 3.** E of different algorithms.

| Image | Original Image | HE | BBHE | CLAHE | BPLHE | RSIHE | ESIHE | MMBEBHE | AICHE |
|---|---|---|---|---|---|---|---|---|---|
| Scene 1 | 6.6795 | 6.2585 | 6.4329 | 7.3721 | 6.6075 | 6.4146 | 6.5399 | 6.4329 | 6.6100 |
| Scene 2 | 6.9335 | 6.6133 | 6.5720 | 7.4798 | 0.8336 | 6.6434 | 6.7688 | 6.5831 | 6.7754 |
| Scene 3 | 6.4832 | 6.0482 | 6.0529 | 7.2479 | 6.1414 | 5.9924 | 6.1180 | 5.9747 | 6.2263 |
| Scene 4 | 5.8144 | 5.7052 | 5.6333 | 6.6725 | 5.7407 | 5.7453 | 5.7456 | 5.6913 | 5.7850 |
| Scene 5 | 5.5746 | 5.0785 | 4.9332 | 6.3464 | 5.0707 | 4.8746 | 5.1037 | 5.0313 | 5.1829 |
| Scene 6 | 6.5849 | 6.2653 | 6.2540 | 7.2336 | 6.3560 | 6.2520 | 6.3487 | 6.2918 | 6.3663 |
| Scene 7 | 7.4980 | 7.2396 | 7.2908 | 7.9450 | 7.4068 | 7.2882 | 7.3781 | 7.3107 | 7.4299 |
| Scene 8 | 6.0384 | 5.1384 | 5.4682 | 6.3106 | 5.5317 | 5.4245 | 5.3574 | 5.7334 | 5.5863 |
| Scene 9 | 7.6653 | 7.4882 | 7.4852 | 7.8956 | 7.5763 | 7.5308 | 7.5508 | 7.4800 | 7.5963 |
| Scene 10 | 6.0335 | 5.0356 | 5.1394 | 6.1589 | 5.2051 | 5.1465 | 5.2067 | 5.1965 | 5.2112 |
| Average value | 6.53053 | 6.08708 | 6.12619 | **7.06624** | 5.64698 | 6.13123 | 6.21177 | 6.17257 | **6.27696** |
| Standard deviation | 0.69256 | 0.86921 | 0.85574 | 0.69431 | 1.89171 | 0.87371 | 0.86821 | 0.81473 | 0.84929 |

Note: The top two averages produced by the compared algorithms are marked in bold font.

**Table 4.** AMBE of different algorithms.

| Image | HE | BBHE | CLAHE | BPLHE | RSIHE | ESIHE | MMBEBHE | AICHE |
|---|---|---|---|---|---|---|---|---|
| Scene 1 | 72.3203 | 8.4028 | 20.4267 | 4.1236 | 3.8192 | 19.3066 | 6.1641 | 1.9737 |
| Scene 2 | 63.5495 | 20.7392 | 22.7322 | 10.680 | 7.7807 | 12.5410 | 13.5527 | 6.6932 |
| Scene 3 | 92.8464 | 24.4811 | 32.1225 | 17.3286 | 18.9348 | 52.7854 | 13.7168 | 12.1253 |
| Scene 4 | 60.7812 | 4.2255 | 12.7312 | 52.0211 | 0.7013 | 31.6171 | 1.6792 | 0.6925 |
| Scene 5 | 68.3038 | 17.6009 | 29.8796 | 42.2594 | 21.6477 | 13.0248 | 12.8289 | 12.6386 |
| Scene 6 | 71.6894 | 30.1702 | 23.7526 | 20.1931 | 17.8950 | 32.4487 | 21.0208 | 17.6621 |
| Scene 7 | 20.4280 | 2.1106 | 20.0978 | 2.4444 | 1.2151 | 0.9639 | 4.3422 | 0.7946 |
| Scene 8 | 67.4038 | 27.0251 | 18.7464 | 55.4875 | 22.5769 | 41.5077 | 64.2064 | 15.7625 |
| Scene 9 | 9.6527 | 5.9391 | 19.8077 | 6.2182 | 5.4492 | 4.6846 | 4.6402 | 4.2561 |
| Scene 10 | 68.8450 | 34.6771 | 47.8343 | 32.3713 | 32.0415 | 26.2767 | 64.6110 | 31.2214 |
| Average value | 59.58201 | 17.53716 | 24.8131 | 24.31272 | 13.20614 | 23.51565 | 20.67623 | **10.382** |
| Standard deviation | 25.12541 | 11.71418 | 9.79701 | 19.97632 | 10.78832 | 16.48196 | 23.76145 | 9.61480 |

Note: The top averages produced by the compared algorithms are marked in bold font.

Based on the evaluation results, the AICHE algorithm proposed herein has good image enhancement effects in all three working conditions; it can protect the average brightness of the image and reduce the merging of gray levels on the basis of improving the image contrast, and it has good robustness. However, as shown in Table 5, the computational time of the AICHE algorithm is relatively long.

**Table 5.** Time of different algorithms.

| Image | HE | BBHE | CLAHE | BPLHE | RSIHE | ESIHE | MMBEBHE | AICHE |
|---|---|---|---|---|---|---|---|---|
| Scene 1 | 3.25 | 4.78 | 10.36 | 4.98 | 2.25 | 3.46 | 4.23 | 7.35 |
| Scene 2 | 2.33 | 3.73 | 11.17 | 4.03 | 2.56 | 1.77 | 3.21 | 7.24 |
| Scene 3 | 2.28 | 6.96 | 11.221 | 7.17 | 1.73 | 2.56 | 6.75 | 8.83 |
| Scene 4 | 3.27 | 5.56 | 10.26 | 5.52 | 3.26 | 3.75 | 4.89 | 7.89 |
| Scene 5 | 2.27 | 6.72 | 9.39 | 6.28 | 1.79 | 2.36 | 4.95 | 7.93 |
| Scene 6 | 1.82 | 3.28 | 11.19 | 3.98 | 2.57 | 2.23 | 3.43 | 6.23 |
| Scene 7 | 4.59 | 6.89 | 13.25 | 5.57 | 3.89 | 4.89 | 5.69 | 10.36 |
| Scene 8 | 1.11 | 4.77 | 9.25 | 6.73 | 1.75 | 1.27 | 5.49 | 7.69 |
| Scene 9 | 1.74 | 4.41 | 11.30 | 3.26 | 2.31 | 1.15 | 2.82 | 5.53 |
| Scene 10 | 2.217 | 6.400 | 11.128 | 7.49 | 2.54 | 2.07 | 4.82 | 8.35 |
| Average value | 2.4877 | 5.35 | 10.8519 | 5.501 | **2.465** | 2.551 | 4.628 | 7.74 |
| Standard deviation | 0.98344 | 1.35159 | 1.38965 | 1.43893 | 0.68844 | 1.16997 | 1.22275 | 1.33549 |

Note: The top averages produced by the compared algorithms are marked in bold font.

## 5. Conclusions

In this study, the AICHE algorithm is proposed for image enhancement under complex working conditions. The problems of average brightness shift, image over-enhancement, and gray level merging in the traditional histogram equalization process are effectively solved. The algorithm improves upon the traditional Otsu method by segmenting the image histogram to solve the problem of mean luminance shift and adaptively obtain the threshold to suppress the gray level to avoid the image over-enhancement phenomenon. Then, it uses the local gray level correction method to avoid the gray level merging problem. According to the experimental analysis, the adaptive local correction method can effectively avoid image over-enhancement and image detail loss, and it can enhance the image contrast and detail information. Compared with other improved algorithms, the AICHE algorithm significantly enhances the PSNR, gray level, and information entropy while avoiding the introduction of noise; its SSIM is closer to 1, and its image visual effect is better.

Due to the pursuit of high-quality measurement accuracy, this method may lead to some defects in time efficiency, which is more suitable for image enhancement in the industrial environment. How to improve time efficiency will be a key issue in future research.

**Author Contributions:** Methodology, B.L. and S.J.; software, B.Y.; resources, S.Y. and D.Z.; writing—original draft preparation, B.Y.; writing—review and editing, B.L. All authors have read and agreed to the published version of the manuscript.

**Funding:** This research was funded by the Guangxi Science and Technology Major Special Project, grant number AA22068064; Guangxi Science and Technology Programs, grant number AD22080042; Guangxi Key R&D Program Projects, grant number AB22035066.

**Institutional Review Board Statement:** Not applicable.

**Informed Consent Statement:** Not applicable.

**Data Availability Statement:** Not applicable.

**Acknowledgments:** We appreciate the support of the Guangxi College Students Innovation and Entrepreneurship Program.

**Conflicts of Interest:** The authors declare no conflict of interest.

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
