# Peer review of "Dual Histogram Equalization Algorithm Based on Adaptive Image Correction"

_applsci, doi:10.3390/app131910649_

Round 1

Reviewer 1 Report

The manuscript “Dual Histogram Equalization Algorithm Based on Adaptive Image Correction” by Bowen Ye et al., propose a dual histogram equalization algorithm based on adaptative image correction (AICHE) which improve previous versions based on the histogram equalization method. Particularly, an enhanced version of the standard two-class Otsu’s algorithm is proposed for histogram segmentation together an adaptive local grayscale correction.  It is demonstrated that the algorithm performs well and is effective for a case of study considering several degrees of illumination.  Before a possible publication, however, the authors should clarify the following points.

1.     How does the algorithm would perform in images where the two features of interest, i.e., foreground and background., are composed by largely different number of pixels? For example, with objects of largely different sizes. This could be problematic during the first histogram segmentation since the classical Otsu’s method fails in case like this. 

It would be interesting to see in the paper an example of applicability of the algorithm in a case like this. This could highlight the potentiality or limitations of the proposed algorithm.

2.     Which was the main motivation to use Otsu algorithm instead of other well-known algorithm, like for example K-means? Author should comment about this in the text.

3.     When mentioning the Otsu’s algorithm please clarify that it refers to the standard two-class Otsu algorithm.

4.     It should be also commented about the limitation of your new procedure. For example, about point (1) and also that is valid only when two features of interest are considered.

Reviewer 2 Report

This paper presents a new dual histogram equalization method called AICHE, aimed at enhancing image contrast for visual measurement of moving arm holes in complex working conditions. The method addresses different issues associated with traditional histogram equalization techniques. The paper also highlights the method's experimental verification and its potential to improve histogram segmentation, contrast, and detail information of the input image.

 The paper is well-written, creative, and sounds scientific. It adds to the body of current knowledge in digital image processing and its real-life applications. While the paper introduces a promising method, several improvements and clarifications are necessary to enhance its quality and comprehensibility:

 1.       Equation Definitions: It's essential to include a "where" clause for all equations to define all variables used. This clarification will improve the understanding of the mathematical aspects of the method, some equations have, and some do not, e.g. equation 18.

 2.       Typos and Consistency: The paper mentions a typo (ALCHE) in Figure 6, which should be corrected to AICHE. And “101g” equation 24. Ensure consistency in spelling throughout the paper, as these typos may appear elsewhere as well, and please check all equations as some have missing information, e.g. in equation 23, the summation should start by i=1 to n or width of image, j=1 to n or height of image. Also, equation 24 used n, and the where clause mentioned N! Or is this where clause belongs to equation 23? To be honest all equations are missy.

 3.       Evaluation Indicators: To make the paper more accessible to a multidisciplinary audience, provide a brief explanation of the evaluation indicators (E, AMBE, SSIM, PSNR) and their respective ranges. Mentioning what constitutes a "good" or "bad" value for each indicator will help readers interpret the results effectively. e.g. high PSNR means good algorithm, and so on.

 4.       Related Work Section: Incorporate a dedicated "Related Work" section to provide context for your research. Additionally, expand the literature review by including references to relevant deep learning approaches (e.g., https://doi.org/10.1007/s10278-022-00721-9) and traditional methods (e.g., 10.1109/ICCIT-144147971.2020.9213752) related to image enhancement and histogram equalization. And identify the research gap that your paper fills. This will demonstrate the paper's contribution to the existing body of research.

 5.       Conclusion and Limitations: Strengthen the conclusion section by summarizing the key findings and contributions of the AICHE algorithm. Address the limitations of the proposed method and suggest possible avenues for future research or improvements.

 6.       Symbol Confusion: It appears that the symbol "E" is used with multiple meanings in the paper. It means average brightness in equation 25, and entropy in equation 24, use different symbols to avoid confusion.

 7.       Experimental Validation: Since the title indicates proposing a general image enhancement method, the paper exhibits a limited experimental scope as it focuses on only one image under various conditions dark, foggy, and dusty. Please conduct experiments on a more diverse set of standard images, including at least 10 samples under various conditions if possible (e.g., dark, foggy, dusty), or without such conditions. Report the average values of the four evaluation indicators along with their standard deviations. Discuss the implications of these results in comparison to existing methods. This is very important to generalize the proposed methods.

 8.       Nothing is mentioned about the time complexity of the method, and the consumed time is never discussed in relation to the other methods.

some typos were identified, and proofreading is needed.

Round 2

Reviewer 2 Report

The authors have addressed all my comments, thanks to them.

minor checks only.